# Emerging Roles of Megakaryocytes in Immune Regulation and Potential Therapeutic Prospects

**DOI:** 10.3390/cells14211677

**Published:** 2025-10-27

**Authors:** Seungjun Kim, Kiwon Lee

**Affiliations:** Department of Bioscience and Biotechnology, Hankuk University of Foreign Studies, Yongin 449-791, Republic of Korea; tmdwns3827@hufs.ac.kr

**Keywords:** megakaryocyte diversity, pluripotent stem cells, immune MK

## Abstract

Megakaryocytes (MKs) have traditionally been viewed as terminal hematopoietic cells responsible solely for platelet production. However, recent advances in imaging and single-cell transcriptomics have revealed substantial heterogeneity among MK populations and diverse functions beyond thrombopoiesis. MKs actively participate in innate and adaptive immunity, modulate the hematopoietic stem cell (HSC) niche, and adapt to physiological and pathological stimuli. Located in distinct anatomical sites such as bone marrow and lung, MKs exhibit compartment-specific specializations that enable them to serve as critical integrators of hemostatic, immune, and regenerative processes. Experimental models using human pluripotent stem cells and inducible MKs have enhanced mechanistic insights, while innovative bioreactor platforms and xenotransplantation strategies advance translational applications in platelet production and therapy. Furthermore, immune MK subsets derived from pluripotent stem cells show promising therapeutic potential for modulating inflammation and autoimmune diseases. Continued exploration of MK diversity, tissue-specific roles, and intercellular communication will unlock new opportunities for leveraging MK plasticity in regenerative medicine, immunotherapy, and hematologic disorders, repositioning these versatile cells as central players in systemic homeostasis and defense.

## 1. Introduction

MKs were first described in the late 19th century and, for much of the 20th century, were considered terminal-stage cells within the hematopoietic hierarchy whose sole role was platelet production [1]. In this classical model, MKs originate from HSCs through megakaryocyte–erythroid progenitors (MEPs), then undergo polyploidization by endomitosis, giving rise to giant, multinucleated MKs with multilobulated nuclei [2,3]. Within mature MKs, the cytoplasmic demarcation membrane system (DMS) forms an extensive membranous network that provides reservoirs for future platelet membranes, subsequent cytoplasmic fragmentation produces circulating platelets [3,4].

For decades, this platelet-centric paradigm dominated the field. However, the advent of advanced experimental tools, particularly intravital microscopy and single-cell RNA sequencing (scRNA-seq), has fundamentally reshaped our understanding of MK biology. Intravital microscopy, including two-photon and multiphoton imaging, has enabled direct visualization of MK dynamics and platelet release in vivo within the bone marrow, lung, and other tissues, with unprecedented spatial and temporal resolution [5,6]. In parallel, scRNA-seq has revealed striking molecular heterogeneity among MK populations across different anatomical sites, such as bone marrow and lung [7,8].

These approaches have uncovered distinct MK subpopulations with specialized functions extending beyond thrombopoiesis, including immune regulation and support of the hematopoietic stem and progenitor cell niche [9,10]. Such findings highlight previously underappreciated immune programs within MKs and demonstrate their roles as cellular integrators at the intersection of hemostasis, immunity, and hematopoietic regulation.

In this review, we summarize the classical view of MK biology. We then highlight recent advances that reveal MK heterogeneity and compartment-specific functions in the bone marrow and lung. We discuss how MKs integrate hematopoietic homeostasis with immune regulation and niche maintenance. Special attention is given to the distinct contributions of bone marrow versus lung MKs, including ongoing debates about their relative roles in platelet production and immune surveillance. We also examine the molecular networks and transcription factors governing MK development, as well as the pathological consequences of their dysregulation. Finally, we review experimental models including pluripotent stem cell–derived MKs, bioreactor systems, in vivo transplantation approaches, and computational models that have advanced mechanistic insights and translational applications. By integrating these perspectives, we underscore the emerging therapeutic potential of MKs, particularly PSC-derived immune MKs, in regenerative medicine, transfusion biology, and immunotherapy.

## 2. Hematopoietic Homeostasis and Platelet Biogenesis

Platelet biogenesis is a highly orchestrated process fundamental to blood clotting and tissue repair. The primary regulator of platelet production is thrombopoietin (TPO), which acts by binding to the MPL receptor (myeloproliferative leukemia protein) expressed on hematopoietic stem cells, MK progenitors, MKs, and platelets. MPL is also expressed on endothelial cells, but typically at lower levels than on platelets, and its physiological role in endothelium is less dominant and primarily relevant in certain disease settings such as cardiovascular dysfunction in myeloproliferative neoplasms [11,12,13]. The TPO/MPL axis plays a central role in MK development, operating in close conjunction with an intricate network of transcription factors that orchestrate lineage commitment and terminal differentiation. Key transcription factors such as RUNX1, GATA1, and FLI1 synergistically regulate megakaryocyte-specific genes, including MPL, reinforcing differentiation and maturation processes [14,15,16] (Figure 1).

Complementing these factors, GATA2 is vital for early progenitor proliferation and MK identity, often compensating when GATA1 is deficient, while also controlling cell cycle progression and repressing myeloid genes [17,18]. The transcription factor TAL1/SCL promotes megakaryocytic lineage specification through cooperation with GATA and ETS family members [19]. NF-E2 is essential for terminal differentiation and platelet biogenesis, and FOG-1 acts as a critical cofactor for GATA1, mediating proper megakaryocyte maturation [19]. Additionally, SP1 modulates cell cycle regulation and maturation and EGR1 contributes to gene expression modulation during differentiation [19] (Table 1). These factors form dynamic multiprotein complexes and enhanceosome-like structures at key gene promoters, enabling extensive transcriptional activation that supports megakaryopoiesis [14]. Epigenetic regulators and microRNAs introduce further layers of control by modifying chromatin and post-transcriptionally regulating key genes, ensuring precise megakaryocyte development [19].

Together, this coordinated network underpins efficient platelet production and hematopoietic homeostasis, highlighting molecular targets for platelet production disorders (Figure 1). Dysregulation of TPO/MPL signaling or transcription factors can cause blood disorders: insufficient TPO/MPL activity or transcription factor mutations lead to thrombocytopenia due to defective MK maturation and platelet production. Conversely, gain-of-function mutations may cause myeloproliferative neoplasms or contribute to acute myeloid leukemia (AML) by promoting abnormal MK proliferation and function [18,31,32] (Figure 1).

RUNX1 produces several isoforms through alternative promoter usage and splicing, most notably RUNX1B and RUNX1C. During megakaryopoiesis, these isoforms have distinct but complementary roles. RUNX1C, primarily expressed in adult hematopoietic stem and progenitor cells, is crucial for both the specification of the megakaryocyte lineage and the maintenance of MK progenitor proliferation and survival [33]. RUNX1B, arising from a different promoter, is particularly important in the initiation and progression of megakaryocytic and erythroid differentiation [34]. Both isoforms collaboratively regulate the expression of critical genes for megakaryocyte maturation and platelet production, such as ITGA2B, MYL9, and PF4 [33,34]. Disruptions in RUNX1 isoform expression or function, as seen in familial platelet disorder with a propensity for myeloid malignancy (FPD/AML), can impair MK differentiation and result in thrombocytopenia and increased leukemia risk. Recent molecular studies highlight that the ratio and regulation of RUNX1 isoforms are tightly controlled and that imbalances may predispose to hematologic disease through impaired megakaryopoiesis and dysregulated gene expression networks [34,35].

In the bone marrow, mature MKs extend proplatelet projections into the sinusoidal vasculature. Blood flow induced shear stress then fragments these proplatelets, yielding 1000–3000 circulating platelets per MK [36,37,38]. This process visualized in real time and confirmed by both in vivo and in vitro studies [39,40]. Recent studies show lung MKs also generate comparable numbers of platelets per MK, supported by unique pulmonary microenvironmental factors in the lung such as adequate oxygenation, healthy pulmonary endothelium, ventilation, and the lung’s microvascular structure, all of which optimize MK fragmentation and platelet release [5,40,41] (Figure 1 and Table 2).

Controversy remains regarding the relative contribution of bone marrow versus lung-MKs to circulating platelet numbers. However, direct comparative establishes that under steady-state conditions, the bone marrow remains the predominant source of circulating platelets, with quantitative analyses revealing bone marrow MKs far outnumber and outperform their lung counterparts in routine hematopoiesis [47]. The lung’s contribution is minor during homeostasis but may increase during stress or thrombocytopenia [40]. Thus, while lung MKs play an important role under specific conditions, bone marrow MKs are the main contributors to total platelet numbers [39,47] (Figure 1 and Table 1). Functional platelets produced in the lung demonstrate typical morphology and responsiveness to activation stimuli, confirming their physiological competence. Mechanistically, interactions between MKs and pulmonary endothelial cells, including signaling via von Willebrand factor (vWF) receptors, further facilitate platelet generation within the lung capillaries [48] (Figure 1 and Table 2).

Importantly, under thrombocytopenic stress, lung MKs increase their platelet output substantially, contributing an estimated 5–10% of circulating platelets under normal conditions, with some studies suggesting contributions as high as 20% depending on physiological or pathological states (Figure 1 and Table 1).

Lung MKs also display distinct lineage markers and immune-related gene expression profiles that differentiate them functionally from bone marrow MKs, highlighting their specialized role in pulmonary immune surveillance in addition to thrombopoiesis [39,40,45], a topic we will discuss in detail later. These findings underscore the lung as a critical, unique site of platelet production where MKs adapt structurally and functionally to dynamic local cues, playing vital roles in systemic platelet homeostasis and immune defense.

## 3. Immune Functions

### 3.1. Immune Receptor Expression

MKs express a broad array of immune receptors, equipping them for both pathogen recognition and immunological communication [9,20,21,22,23]. Notably, MKs display multiple Toll-like receptors (TLR1–6), including TLR5 expressed at the mRNA level in lung MKs, enabling detection of pathogen- and damage-associated molecular patterns through activation of downstream NF-κB and MAPK signaling pathways that drive pro-inflammatory responses [21]. MKs also express NOD-like receptors and C-type lectins that contribute to sensing intracellular pathogens and mediating endocytic uptake. Regarding Fc receptors, MKs express human FcγRIIA, a low affinity activating receptor shared with platelets whereas murine MKs express high affinity FcγRI on a subpopulation; stimulation of FcγRI by ligands such as C-reactive protein potentiates MK microparticle production [21]. Additionally, MKs express FcεRI, MHC class I and II, and co-stimulatory molecules including CD80 and CD86, facilitating antigen processing and presentation to T cells. While MHC class II is downregulated during MK maturation, it persists in select subpopulations, allowing MKs to participate in adaptive immune responses and potentially prime T cells within hematopoietic and peripheral tissues. These receptors empower MKs to take an active role in both innate and adaptive immunity, facilitating their direct involvement in host defense and intercellular communication in hematopoietic and peripheral tissues [9,20,21,22,23] (Figure 1 and Table 1).

### 3.2. Cytokine and Chemokine Secretion

MKs produce and secrete a broad spectrum of cytokines and chemokines, including IL-6, TNF-α, TGF-β, and notably CXCL4 (PF4), which play critical roles in orchestrating immune cell recruitment, activation, and differentiation [20,21,49]. CXCL4 stands out for its dual function, acting as a chemotactic factor that attracts immune cells while also exerting immune-suppressive effects, helping maintain immune balance. IL-6, TNF-α, and TGF-β contribute to inflammation and immunoregulation by modulating the activity and phenotype of various immune cells (Table 1).

These signaling molecules are stored in MK α-granules or released via microparticles, enabling MKs to influence both the hematopoietic niche and peripheral immune responses by shaping the local microenvironment and controlling stem cell behavior [20,50,51]. Molecules stored in α-granules undergo release through fusion of the α-granule membrane with the plasma membrane or the open canalicular system, a process mediated by SNARE proteins such as VAMP-8 and syntaxins that coordinate membrane docking and fusion to secrete granular contents extracellularly [24] (Table 1). This exocytosis releases bioactive proteins and increases platelet surface area to facilitate cellular interactions. In addition to direct exocytosis, MKs release signaling molecules as microparticles small extracellular vesicles formed by plasma membrane budding and blebbing involving cytoskeletal remodeling. These microparticles carry proteins, lipids, and nucleic acids that modulate both local and distant hematopoietic and immune environments, expanding MK influence beyond their immediate niche [24,25,26,27].

For example, MK-derived TGF-β helps maintain HSC quiescence and modulates differentiation by signaling through TGF-β receptors that activate SMAD-dependent transcriptional programs, thereby promoting HSC dormancy and preventing exhaustion [28]. CXCL4, a platelet-derived CXC chemokine, impacts immune cell migration and survival by binding to receptors such as CXCR3 and modulating the recruitment and polarization of monocytes and macrophages. Specifically, CXCL4 promotes monocyte survival and skews macrophage differentiation toward a unique phenotype distinct from classical M1 or M2 profiles, influencing inflammatory responses and tissue remodeling [29].

Additionally, CXCL4 modulates the extracellular matrix and interacts with proteoglycans, thereby facilitating widespread immune cell recruitment independent of traditional chemokine receptor signaling [30]. These molecular activities enable MK-derived factors like TGF-β and CXCL4 to fine-tune both the hematopoietic and immune microenvironments (Figure 1 and Table 1). Additionally, MKs interact with mesenchymal stromal cells and endothelial cells in the bone marrow niche, cooperating to regulate hematopoiesis through cytokine networks, including IL-6, IL-11, stem cell factor (SCF), and others. Beyond steady-state regulation, MK cytokines may be particularly important in emergency hematopoiesis during inflammation or infection, supporting rapid immune and platelet responses (Table 1). This cytokine and chemokine repertoire underpins MKs’ versatile roles at the intersection of hemostasis, immunity, and hematopoietic niche maintenance.

### 3.3. Direct Immune Activities

Select MK subsets exhibit upregulated antigen-presenting capabilities, particularly through expression of MHC class II molecules and antigen processing machinery, primarily based on mouse model studies. Recent studies have shown that about 20–30% of MKs in murine bone marrow express MHC class II, especially in platelet-biased progenitors and mature MKs, with expression increasing under certain conditions such as aging. These MKs also display co-stimulatory molecules like CD80, CD86, and CD40, which are essential for effective T cell activation (Table 1). Functionally, bone marrow-derived MKs can present antigens to CD4^+^ T cells, supporting adaptive immune activation, and this antigen-presentation capacity is enhanced following Toll-like receptor (TLR) stimulation. Human MKs derived from bone marrow CD34^+^ cells similarly upregulate MHC class II pathways, suggesting a conserved immunoregulatory across species. Human megakaryocyte progenitors, particularly those at immature stages, express MHC class II and function as professional antigen-presenting cells capable of activating CD4^+^ T cells and modulating adaptive immunity. This expression diminishes with MK maturation, consistent with observations in mice [10,52].

MHC class I molecules are broadly expressed across most MK subsets and are primarily involved in presenting endogenous antigens to CD8^+^ T cells. Mature MKs possess the full antigen processing machinery to load peptides onto MHC class I molecules, and these antigen-MHC class I complexes can be transferred from MKs to proplatelets, which then pass them to circulating platelets. This enables platelets to actively participate in antigen cross-presentation and immune surveillance. MHC class II expression is more restricted, detected in approximately 20–30% of bone marrow MKs, particularly enriched in platelet-biased progenitors and a subset of immune-oriented small, low-ploidy MKs. These MHC class II+ MK compartments exhibit upregulated antigen processing pathways, allowing presentation of exogenous antigens to CD4^+^ T cells. Conditions such as aging and TLR stimulation increase MHC class II expression in these subsets, reflecting an inducible immunoregulatory role [10,43,53].

Co-stimulatory molecules, including CD80, CD86, and CD40, which are crucial for effective T cell activation, are expressed not only on MHC class II-expressing MKs but also across various MK subsets that lack MHC class II [10,43,53]. This wider distribution implies that multiple MK compartments can modulate immune responses, potentially via different mechanisms beyond classical antigen presentation [10].

Further molecular insights reveal that mature MKs which are MHC class II-negative still retain MHC class I expression and can efficiently cross-present endogenous antigens to CD8^+^ T cells, highlighting their role beyond platelet formation into immune surveillance [53]. Additionally, single-cell transcriptomic analyses identify distinct MK progenitor subsets characterized by differential expression of markers such as CD24a, von Willebrand factor (VWF), and CD48 [10]. These markers correlate with varying capacities for antigen presentation and immune modulation, indicating a complex hierarchy in MK lineage commitment and function. Moreover, unique ‘immune MK’ populations reside in peripheral organs, such as the lung, that express higher levels of immune-related molecules compared to bone marrow MKs [45]. These cells function in pathogen clearance and robust antigen presentation via MHC class II to CD4^+^ T cells, emphasizing tissue-specific specialization within MK compartments [43]. This detailed molecular and cellular compartmentalization of MHC class I and II expression, alongside co-stimulatory molecule distribution, illustrates the multifaceted and conserved immunoregulatory roles of megakaryocyte subsets in orchestrating adaptive immune responses.

A unique MK process called emperipolesis involves the engulfment and transit of neutrophils and other hematopoietic cells through MKs, facilitating bidirectional molecular exchange that influences both platelet production and immunomodulation [53,54]. Beyond antigen presentation, MKs exhibit phagocytic activity and, upon activation, can release extracellular chromatin webs resembling neutrophil extracellular traps (NETs), which entrap pathogens and regulate inflammation during infections and sepsis [55,56] (Figure 1). Lung MKs are molecularly distinct with elevated immune gene expression; they internalize and process antigens, effectively present them via MHC class II, and potently activate CD4^+^ T cells, further highlighting the specialized immune regulatory roles of MK subsets in different tissues [57] (Table 2).

Together, these findings reveal that MKs serve not only as platelet producers but also as active players in adaptive immunity, bridging innate and adaptive responses through antigen presentation, phagocytosis, and immune modulation.

### 3.4. Lung MK Specialization

Lung MKs are uniquely adapted both molecularly and anatomically for immune surveillance within the pulmonary environment. These MKs possess a distinct transcriptional profile enriched in immune-related gene expression, setting them apart from their bone marrow counterparts [7,43]. These lung MKs express higher levels of pattern recognition receptors such as Toll-like receptors (e.g., TLR2, TLR4), MHC class II molecules, and antigen presentation-related molecules like CD74 (Table 2). Additionally, they express markers typical of tissue-resident leukocytes and antigen-presenting cells, reflecting their specialized immunological function in microbial surveillance, antigen presentation, and immune regulation.

Strategically located in the lung capillaries and interstitial spaces, lung MKs directly engage with circulating immune cells, enabling rapid response to blood-borne pathogens or pulmonary insults [7,10,42]. They execute multiple immune functions, including antigen processing and presentation, secretion of inflammatory mediators such as cytokines and chemokines, and phagocytosis of inhaled particles. These activities position lung MKs as frontline sentinels at the lung-blood barrier, orchestrating immune cell activation and trafficking to protect lung tissue from infection and damage [10,23,44].

In addition to these roles, lung MKs exhibit responsiveness to pulmonary inflammation and injury, with enhanced secretion of inflammatory molecules like CXCL1, TNF-α, IL-1α, and others upon stimulation. They support immune cell migration, bacterial clearance, and contribute to tissue remodeling. Notably, lung MKs contribute to platelet production locally, with platelets showing immune-modulatory properties that further influence lung immunity [7,44] (Table 1 and Table 2).

Most intriguingly, lung MKs adopt an “immune phenotype” even during fetal development independent of microbial exposure, implying that the lung microenvironment intrinsically promotes their immunological specialization. This positions lung MKs as critical integrators of hemostasis and immunity, ensuring protective surveillance and rapid response in the frontline respiratory tissue [7,44].

## 4. Hematopoietic Niche Maintenance

In the bone marrow, MKs are strategically localized adjacent to arteriolar HSC niches where they play a crucial regulatory role through both secreted factors and direct cell interactions [46,58,59]. MK-derived cytokines such as TGF-β1 and CXCL4 maintain HSC quiescence and niche stability during normal homeostasis. These factors help keep the stem cells in a dormant, non-proliferative state, which is essential for preserving their long-term regenerative capacity and preventing stem cell exhaustion (Figure 1). During hematopoietic stress or injury caused by events like chemotherapy, radiation, or infection, MKs dynamically shift their secretory profile to produce regenerative growth factors including FGF1, IGF-1, and VEGF (Table 1). These factors stimulate HSC proliferation and recovery, enabling rapid restoration of blood cell production. MKs also contribute to niche remodeling under stress, facilitating expansion of supportive osteoblasts and endothelial cells while coordinating repair processes (Figure 1).

The crosstalk between MKs and other niche components is central to their regulatory function. MKs interact intimately with endothelial and mesenchymal stromal cells through adhesion molecules (e.g., ICAM-1, VCAM-1) and chemokine signaling, with the CXCL12-CXCR4 axis playing a crucial role in guiding MK localization, maturation, and stem cell migration. This interplay ensures coordinated regulation of platelet release and HSC maintenance. MKs also produce extracellular matrix proteins such as fibronectin, laminin, and collagen, which contribute to niche structural integrity and signaling [26,46,60,61,62].

Moreover, aberrant MK activation can disrupt normal niche function and contribute to pathology. In myeloproliferative diseases such as myelofibrosis, excessive secretion of profibrotic factors like platelet-derived growth factor (PDGF) and TGF-β by MKs drives pathological fibrosis within the bone marrow, impairing hematopoiesis and niche function (Table 1). MKs closely interact with endothelial and mesenchymal stromal cells in the vascular niche, engaging in complex cellular crosstalk via adhesion molecules and chemokine signaling (e.g., CXCL12-CXCR4 axis) that coordinate MK maturation, platelet release, and stem cell niche regulation. This spatial proximity and signaling integration position MKs as pivotal gatekeepers of the hematopoietic microenvironment, tailoring stem cell behavior and niche remodeling depending on physiological or pathological cues [46,59].

Overall, MKs serve as pivotal gatekeepers within the bone marrow niche, finely tuning the balance between HSC preservation and activation through spatial positioning, molecular signaling, and dynamic functional shifts to meet systemic hematopoietic demands.

## 5. The Bridge

MKs serve as a vital bridge between hemostasis and immunity, utilizing their anatomical niches and functional versatility to coordinate systemic homeostasis and defense. Rather than attributing direct “hemostatic function” to MKs, which is primarily executed by platelets, we emphasize MKs as upstream regulators that enable platelet production while simultaneously shaping immune and niche environments. In steady-state conditions, MKs continuously produce platelets for hemostasis and support HSC quiescence and niche stability through secretion of cytokines, particularly CXCL4 and TGF-β. These factors maintain stem cell dormancy, regulate differentiation trajectories, and preserve long-term regenerative capacity, ensuring hematopoietic balance. Through spatial positioning near arteriolar HSC niches and physical interactions with stromal and endothelial cells, MKs directly contribute to structural and molecular regulation of the hematopoietic microenvironment [9,36,50,56]. Upon injury, infection, or thrombocytopenic stress, MKs undergo a functional shift resembling the broader concepts of “immunothrombosis” and “thromboinflammation,” where clotting and immunity converge.

However, MKs act not as executors of clotting but as upstream coordinators that shape both platelet output and immune signaling. In this state, MKs upregulate expression of immune receptors, including TLRs, Fcγ receptors, and antigen-presentation molecules such as MHC class II, co-stimulatory molecules such as CD40, CD80, and adhesion molecules such as ICAM-1 and VCAM-1. They secrete inflammatory cytokines such as IL-1α, TNF-α, IL-6, and chemokines including CXCL1 and CCL2, which mobilize neutrophils, monocytes, and lymphocytes. Through antigen presentation and cytokine-driven recruitment, MKs amplify adaptive immune responses and participate in shaping the inflammatory microenvironment.

In addition, MKs release regenerative growth factors including FGF1, IGF-1, and VEGF that stimulate hematopoietic recovery and vascular remodeling, ensuring rapid restoration of hematopoiesis following stress. These dual roles, immune regulation and regenerative support, demonstrate how MKs function as integrators, balancing host defense and tissue repair [9,41,56].

Lung MKs, strategically positioned within pulmonary capillary beds, further highlight this bridging role. They rapidly replenish platelets in circulation under stress conditions while also serving as frontline immune sentinels. Their heightened immune gene expression and ability to process and present antigens uniquely equip them to detect pathogens in the lung vasculature, coordinate local immune surveillance, and contribute to lung tissue protection. By contrast, bone marrow MKs focus more on sustaining the HSC niche and ensuring systemic hematopoietic stability [39,40,41,44,45,48].

Thus, MKs should not be defined by platelet-centered hemostasis alone. Instead, they are versatile regulators that integrate platelet biogenesis, immune signaling, and stem cell niche dynamics in a tissue- and context-specific manner. By reframing MKs as “bridges” rather than direct effectors of hemostasis, our review emphasizes their upstream, regulatory roles that connect vascular biology, immunity, and hematopoietic maintenance (Figure 1).

## 6. Experimental Models and Translational Implications

Advances in stem cell technologies enable differentiation of human HSCs and induced pluripotent stem cells (iPSCs) into MKs. These models allow for precise genetic engineering, disease modeling (such as RUNX1 mutations in familial platelet disorder with a propensity for myeloid malignancy), and high-throughput studies of MK biology [63,64]. hPSCs and iPSC-derived MKs recapitulate critical genetic and transcriptomic characteristics of native MKs, facilitating robust functional analyses [65,66].

Bioreactor platforms have emerged as pivotal innovations, using physiological parameters such as oxygen tension (10–30% pO_2_) and controlled shear stress (100–400 µL/min) within three-dimensional matrices to improve in vitro platelet release from megakaryocytes (MKs) and achieve scales suitable for clinical transfusion [67,68,69]. Innovative microfluidic systems applying controlled shear to MKs seeded on von Willebrand factor (vWF)-coated surfaces have accelerated platelet production, with released platelets analyzed for producing multiple platelets per MK. These platelets exhibit functional properties such as aggregation and expression of CD41 and CD42b comparable to donor platelets. Bioreactor modules efficiently simulate vascular niche conditions, enabling rapid and efficient platelet shedding within hours, as opposed to the slower maturation in static cultures, significantly faster than traditional static cultures. Such advances hold promise for scalable, high-quality platelet manufacture to address donor shortages and advance transfusion medicine [63,67,68,69,70,71]. These systems optimize both quantity and quality of platelet production, replicating vascular environments and enabling scalable generation for therapeutic use.

Contemporary bioreactor designs incorporate integrated microfluidic platforms with features including bone marrow vascular niche architecture, temperature control, gas exchange, continuous process flow, and application of precise vascular shear stress [69,71,72,73]. The unique designs enable selective trapping of megakaryocyte progenitors and real-time visualization of platelet production. Transitioning from 2D to 3D designs and introducing porous membranes and increased functional surface areas have fostered increased platelet yields. Commercial scalability remains a challenge but ongoing improvements in automation, computational fluid dynamics modeling, biocompatible materials, and media management are anticipated to further augment production efficiency and reduce costs [69,71,72,73].

Moreover, platelet-based therapeutic applications increasingly harness both intact platelets and platelet derivatives for targeted drug delivery and immunomodulation. Engineered platelets and platelet membrane–coated nanoparticles provide innovative drug delivery platforms, enhancing the precision and efficacy of chemotherapy, immunotherapy, and gene therapy by leveraging platelets’ inherent ability to home to sites of vascular injury, inflammation, and tumor microenvironments [74,75,76]. Platelet-derived membranes cloak nanoparticles with “self” identity, prolonging circulation time and evading immune clearance while delivering small molecules, antibodies, or nucleic acids directly to pathological sites [74,77]. In oncology, platelet-coupled strategies have been shown to deliver checkpoint inhibitors and oncolytic agents selectively to tumor vasculature, thereby amplifying local immune responses and reducing systemic toxicity [76,78]. Similarly, platelet extracellular vesicles (PEVs), including microparticles and exosomes, have emerged as natural nanocarriers capable of traversing biological barriers such as the blood–brain barrier, expanding therapeutic opportunities in neuroinflammation and brain tumors [77,79]. Beyond drug delivery, engineered platelets are being developed as active immunomodulatory agents, capable of presenting tumor antigens or delivering cytokines to boost local immune activation [76].

Importantly, advances in large-scale production of functional platelets from human pluripotent stem cells (hPSCs) and bioreactor systems ensure the feasibility of “off-the-shelf” platelet-based therapeutics, paving the way for broad clinical applications in cancer therapy, cardiovascular repair, autoimmune disease modulation, and regenerative medicine [69,73]. Together, these innovations highlight the dual utility of platelets as both natural effectors and engineered vehicles, positioning them at the forefront of next-generation targeted therapies. Transplantation of human CD34^+^ progenitor–derived MKs into immunocompromised mice enables in vivo study of human MK development and platelet production in a physiological setting. For example, humanized NOD/SCID mice grafted with human CD34^+^ cells generate functional MKs and platelets, with platelet levels improved by macrophage depletion to prevent clearance [80]. Similarly, xenotransplantation of mobilized human CD34^+^ cells in mice yields circulating, functional human platelets, providing a robust model for studying human thrombopoiesis and testing therapies [65,66,81]. These models offer valuable platforms for translational research. Animal models such as murine xenotransplantation and nonhuman primate studies help validate in vivo relevance and clinical potential of engineered MKs, including assessing immunogenicity, long-term engraftment, and thrombopoietic capacity of human platelets in complex physiological environments. These models are essential for bridging in vitro data to patient therapies [82,83,84,85].

Computational modeling has become a central tool in optimizing MK differentiation and artificial platelet production. Recent studies, such as Garzon Dasgupta et al., have demonstrated that advanced bioreactor systems including Taylor-Couette devices use computational fluid dynamics (CFD) simulations to precisely control shear forces and turbulence applied to individual MKs [86]. These models are used to design bioreactors that maximize platelet yield by ensuring uniform mechanical stimulation across all MKs in culture. CFD and in silico simulations (performed with software like ANSYS Fluent 2025 R2) enable prediction of flow regimes, optimization of reactor geometry and rotation speeds, and validation of scalability for industrial-level production. The ability to maintain uniform impact on MKs, rather than merely providing a generalized turbulent environment, was critical for efficient, large-scale platelet release [86]. Beyond fluid mechanics, mathematical frameworks based on differential equations are employed to simulate cellular lineage progression and differentiation outcomes, allowing researchers to model the kinetics of megakaryopoiesis, optimize cytokine stimulation protocols, and predict cell yield under various conditions. These models are instrumental for mapping out the entire production process from stem cell expansion to MK commitment and terminal platelet formation [87].

Data-driven and computational approaches are also essential for scaling up bioprocesses, setting process parameters, and assessing biosafety. For instance, they guide the selection of optimal protocol variables (such as cell seeding density, cytokine dosing schedules, and oxygen tension), reducing the need for extensive empirical testing during protocol development and bioprocess transfer to manufacturing. In parallel, computational models are increasingly used to anticipate and mitigate risks in cell engineering and to evaluate the efficiency of genetic modifications or pharmacological treatments on MK differentiation pathways [86,87]. Furthermore, recent translational studies demonstrate the therapeutic potential of immune MKs derived from hPSCs in inflammation and autoimmunity. For example, treatment of PSC-derived MKs with P2Y12 inhibitors has been shown to reduce interferon-alpha (IFNα) signaling pathways, which are known to play a key role in systemic lupus erythematosus (SLE), a severe autoimmune condition. RNA sequencing analysis revealed that P2Y12 inhibition downregulates gene expression related to inflammatory and interferon-associated responses in both MKs and platelets. Functionally, this leads to decreased inflammatory platelet-leukocyte interactions, which are important contributors to disease pathogenesis in SLE [88,89]. This suggests that engineered or pharmacologically modulated immune MKs could help control immune-mediated inflammation. Additional studies support using immune MKs to modulate cytokine release and cell interactions, highlighting their promise for targeted therapies in autoimmune and inflammatory disorders [90].

## 7. Conclusions

MKs, once viewed solely as platelet-producing end-stage cells, are now recognized as versatile regulators at the intersection of hemostasis, immunity, and hematopoietic niche maintenance. Advances in imaging and single-cell transcriptomics have revealed substantial molecular heterogeneity among MK populations, highlighting compartment-specific specialization and previously underappreciated immune functions. MKs not only sustain platelet biogenesis but also actively participate in innate and adaptive immunity, modulate the hematopoietic stem cell niche, and respond dynamically to physiological and pathological cues.

Experimental models leveraging human pluripotent stem cells and inducible megakaryocytes have transformed our ability to dissect MK biology with precision, enabling genetic engineering, disease modeling, and high-throughput functional studies. Innovative bioreactor platforms simulate vascular shear stress to enhance scalable platelet production, addressing a critical need in transfusion medicine. Furthermore, xenotransplantation of human progenitor-derived MKs into immunocompromised mice provides vital in vivo systems to study human thrombopoiesis and therapeutic efficacy.

Translational research emphasizes the therapeutic potential of immune MK subsets, particularly those derived from pluripotent stem cells, in controlling inflammation and autoimmune disorders. Pharmacologic modulation demonstrates promise in attenuating pathogenic immune signaling pathways, underscoring MKs as targets and tools for novel immunotherapies.

Future research aimed at refining our understanding of MK diversity, tissue-specific functions, and intercellular communication will further illuminate their roles in health and disease. Harnessing MK plasticity promises novel therapeutic avenues in regenerative medicine, immunotherapy, and beyond, positioning these once-overlooked cells as pivotal players in systemic homeostasis and defense.

## Figures and Tables

**Figure 1 cells-14-01677-f001:**
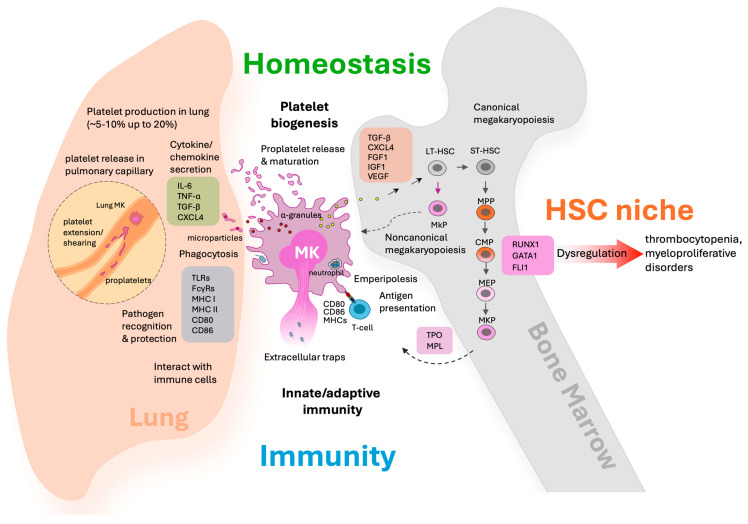
Megakaryocytes as multifunctional regulators in hematopoietic homeostasis and immune surveillance across lung and bone marrow compartments. This schematic illustrates the diverse roles of megakaryocytes in the lung and bone marrow environments. In the lung, proplatelet elongation and shear forces in pulmonary capillaries facilitate platelet release (highlighted with dashed line). Also, MKs interact with immune cells such as contributing to inflammation, pathogen recognition, phagocytosis, and the formation of extracellular traps. These activities highlight MK contributions to innate/adaptive immunity and immune surveillance. MKs perform emperipolesis (engulfment of other cells such as neutrophils), present antigens via MHC class II to T cells, and modulate inflammatory responses. In the bone marrow, MKs support HSC maintenance through noncanonical pathways and secreted factors, further bridging the processes of thrombopoiesis and immune regulation. The diagram emphasizes the dual compartmental roles of MKs, integrating platelet production, immune surveillance, HSC niche support, and their function as a cellular bridge between hemostasis and immunity.

**Table 1 cells-14-01677-t001:** Functional roles and key molecular mediators of megakaryocytes in hemostasis, immunity, and hematopoietic niche regulation.

Category	Functions	Key Factors	Notes	References
**Hematopoietic** **Homeostasis**	MK maturation/ Platelet biogenesis	GATA2, TAL1 (SCL), SP1, EGR1 (early progenitor, lineage specification, maturation) Thrombopoietin (TPO)–MPL axis, RUNX1, GATA1, FOG1, NF-E2, FLI1	Drive MK differentiation and maturation	[9,17,18,19,20,21,22]
	Proplatelet formation	Demarcation membrane system (DMS)	Provides membrane reservoirs for platelet generation	[20,22,23]
	Platelet release	Shear stress in bone marrow sinusoids & lung capillaries	1000–3000 platelets per MK (bone marrow); up to ~20% platelet contribution in lung	[20,21]
**Immunity**	Pathogen recognition	TLR1–6, NOD-like receptors, C-type lectins	Enable MKs to sense Pathogen-Associated Molecular Patterns (PAMPs)/Damage-Associated Molecular Patterns (DAMPs)	[9,21,22]
	Antigen presentation	MHC class I & II, CD80, CD86, CD40	Some subsets present antigens to CD4^+^ T cells	[9,21,22,23]
	Cytokine & chemokine secretion/Granule exocytosis, microparticles	IL-6, TNF-α, TGF-β, CXCL4 (PF4) VAMP-8/syntaxin-mediated release microparticles carry proteins/lipids/nucleic acids	Modulate immune cell recruitment and activation	[9,21,22,24,25,26,27,28,29,30]
	Direct immune defense	Phagocytosis, extracellular chromatin webs (NET-like), emperipolesis	Entrap pathogens; cell–cell material exchange with neutrophils	[9,21]
	Lung-specific immune surveillance	High immune gene expression, antigen processing, phagocytosis, CXCL1, TNF-α, IL-1 α	Lung MKs act as frontline sentinels in pulmonary capillaries enhancing secretion of cytokines, immune cell migration, contributing to tissue remodeling	[21,22]
**Niche** **regulation**	HSC maintenance	CXCL4 (PF4), TGF-β1	Maintain HSC quiescence and stability	[20,22,23]
	Regeneration under stress	FGF1, IGF-1, VEGF	Promote HSC proliferation and recovery	[21,23]
	Pathological remodeling	PDGF, excessive TGF-β	Drive fibrosis in diseases like myelofibrosis	[21,22]
	Anatomical positioning	Localization near arteriolar HSC niches	Enable direct regulation of hematopoietic stem/ progenitor cells	[21,22]
	Crosstalk with stromal/endothelial cells	IL-6, IL-11, SCF	Cooperate with niche cells via cytokine networks; critical in emergency hematopoiesis	[9,20,21,22]

**Table 2 cells-14-01677-t002:** Comparative phenotypic and functional features of lung- and BM-MKs.

Feature	Lung MKs	BM MKs	References
**Lifespan/** **Origin**	Longer-lived (up to months in mouse models) May be seeded from BM HSPCs but also possible lung-resident HSPCs	Short-lived (days to a week) Arise from BM HSPCs exclusively	[10,40,42]
**Maturity/** **Proliferation**	Higher proportion of mature MKs High ploidy Higher expression of maturation genes (*Tubb1*, *Gp1ba*) Lower enrichment for proliferation	Mixed, typically lower average ploidy Variable; ABM-1 cluster shows high *Tubb1*, *Gp1ba*, but proportions of mature MKs are lower Some clusters (ABM-2, ABM-3) enriched for cell division/DNA processing genes (immature profile)	[3,7,10,17]
**Immunity/** **Inflammation**	Enriched for immune-related signatures; higher *TLR2*, *TLR4*, *MHCII*, *Cd74*, *CCR7* antigen presentation genes chemokines	Less immune marker enrichment; lower TLRs, MHCII, and chemokine expression Lower levels of antigen presentation genes	[21,43,44]
**Platelet** **Production** **Potential**	Skewed toward efficient platelet production due to high maturation marker expression	BM ABM-1 is likewise skewed, but with overall lower proportions of terminally mature MKs	[7,17,40,44]
**Environmental** **Phenotype** **Plasticity**	BM MKs acquire immune phenotype after lung residency	MKL retain/express high immune markers in lung environment	[42,44,45,46]

## Data Availability

Not applicable.

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
