# Peer review of "Emerging Roles of Megakaryocytes in Immune Regulation and Potential Therapeutic Prospects"

_cells, 2025, doi:10.3390/cells14211677_

Round 1

Reviewer 1 Report

Comments and Suggestions for Authors

This paper addresses new roles of an interesting and important cell, the megakaryocyte. The authors discuss both the canonical bone marrow megakaryocyte as well as the more recently investigated lung-derived megakaryocyte.

While the article is generally sound, the article contains much repetition and would be greatly enhanced by a careful revision. Additionally, very little mechanistic detail is provided and increased molecular detail would be enhance the paper. Presently,  the same handful of molecules are discussed in each section, and the work appears superficial. Throughout, there is a tendency to focus on a few molecules and these are mentioned repeatedly. Megakaryocyte generation and function are extremely complex and insufficient detail is presented. The table provided needs reformatting and revision.

Specific comments:

1) Title: There is very little info provided on therapeutics with pluripotent stem cell-derived MKs. Either expand this section or alter the title.

2) line 64. MPL expressed as well on platelets and to a lesser extent, endothelial cells.

3) several relevant megakaryocyte specific genes were not mentioned at all: have included GATA-2, SCL, SP-1, NF-E2, EGR1, FOG1, among others….Diffs lung vs BM. To what extent are lung cells sourced in BM?

4) Please address controversy regarding relative contribution of  MKs from lung vs BM to total circulation platelet numbers (see PMID: 37879046).

6) line 109. Livada 2024 paper is suggesting up to 20% not 50% circulating lung-derived platelets.

7)  lines 69-73 Q; are there any diseases specifically related to over or under localization of lung MKs?

Table 1.

8) Please add references for each line.

9) The formatting needs improvement so that words are not broken up.

10) Words are poorly positioned rendering it difficult to see which line goes with which

11) entries should be reviewed to ensure that appropriate descriptions are used. -e.g. Funcn :Proplatelet formation: Only DMS is mentioned as a key factor, whereas a large number of other molecular and cytoskeletal mediators are involved.

12) Platelet release line: adjust 50% to 20% .

13) Section 3.2 . To increase detail, consider adding cognate receptor, and cell types on which found.

  1. Line 138. Clarify that molecules stored in a-granules undergo release through fusion with the plasma membrane and specify how they are released as microparticles.

15) Line 152. Please provide some detail as to MK compartments  harboring MHC Classi vs Class II receptors. n.B. CD 80, and other co-stimulatory molecules can also be found on multiple MK subsets, even not expressing MHC Class Ii.

16) LINE 1160. Clarify that previous section was based on mouse studies (not stated currently).

17) Lines 181/182.Provide some examples.

18) line 224. Font is different.

19) line 231. Explain mechanism.

Reviewer 2 Report

Comments and Suggestions for Authors

Although this review is well-organized and wrote, there are many similar reviews, such as PMID: 40710306, PMID: 40145277, PMID: 39731484, MID: 39542989. Can you describe the differences of your review with these published reviews.

The pages of References 22, 25, 36, 37, 42, and 46 are lacked.

Reviewer 3 Report

Comments and Suggestions for Authors

This article briefly summarized findings of megakaryocytes (MKs) in regulating immunity and the hematopoietic stem cell (HSC) niche under physiological and pathological conditions.

This article reviewed critical and novel functions of MKs, and discussed the compartment-specific, i.e., bone marrow- and lung-located MKs, specializations of MKs. The summarization will benefit the study of MKs/platelets in immunity and hematopoiesis.   

Comments:

  1. Title for Section 2 “hemostasis and platelet biogenesis”. Whether it should be “Homeostasis and biogenesis of platelets”? Also, whether “hemostasis ” in Table 1 should be changed to “Homeostasis”? In this section, the role of shear stress in platelet generation by MKs in BM and lungs should be described.
  2. In section 3, the immune function of MKs in bone marrow and lung should be distinguished better in the presentation.
  3. References should be provided for Description in Table 1, meaning references should be added after the conclusion or description in Table.
  4. Section 4 is too simplified, and more details for the niche regulation by MKs should be provided. For instance, how MKs affect stem cells/progenitor cells, stomal cells, and blood endothelium, etc.
  5. The concept of “bridge” in section 5 is not clear. Whether the “Bridge” is similar to immune-thrombosis and thrombo-inflammation of platelets? In addition, we often state that hemostasis function of platelets, but not hemostasis function of MKs because it is indirect function of MKs. Thus, it would be better to change the related wording.
  6. In section 6, knowledge of animal models and potential application of computational models should be included.

Round 2

Reviewer 1 Report

Comments and Suggestions for Authors

The authors have greatly improved their manuscript. There are just a few small comments:

1) The max production was updated in the text, but Figure 1 is still stating platelet production in the lung as 50%.

2) Ref 30 is missing.

3) p. 10, lines 360-364. All references are review articles. Please provide at least one primary reference.

4) p.12, par. 2. Re:ref. 90. The Taylor-Couette device is not a computational fluid dynamic simulation system as implied in the text. It is a physical device which was used to enable tracking of particles/cells under very precise fluid dynamic parameters, in the presence/absence of RBC to drive blood vessel type flow and cellular collisions. Please clarify in the text.
